



# Added value of the EURO-CORDEX high-resolution downscaling over the Iberian Peninsula revisited. Part I: Precipitation

João António Martins Careto[1], Pedro Miguel Matos Soares[1], Rita Margarida Cardoso[1], Sixto Herrera[2] and Jose Manuel Gutiérrez[3]

[1]Instituto Dom Luiz, Faculdade de Ciências, Universidade de Lisboa, 1749-016, Portugal

[2]Meteorology Group. Dept. of Applied Mathematics and Computer Science. Universidad de Cantabria. Santander, Spain.

[3]Meteorology Group. Instituto de Física de Cantabria, CSIC-University of Cantabria, Santander, Spain.

*Corresponding author João A. M. Careto (jacareto@fc.ul.pt)*

*Faculdade de Ciências da Universidade de Lisboa, Campo Grande, Ed. C8 (3.01)*

*1749-016 Lisboa, Portugal*





**Abstract.** Over the years higher resolution Regional Climate Model simulations have emerged owing to the large increase in
computational resources. The 12 Km resolution from the Coordinated Regional Climate Downscaling Experiment for the European
domain (EURO-CORDEX) is a reference, which includes a larger multi-model ensemble at a continental scale while spanning at
least a 130-year period. These simulations are computationally demanding but not always revealing added value. In this study, a
recently developed regular gridded dataset (Iberia0.1) and a new metric for added value quantification, the distribution added value
(DAV), are used to assess the precipitation of all available EURO-CORDEX Hindcast (1989-2008) and Historical (1971-2005)
simulations. This approach enables a direct assessment between the higher resolution regional model runs against their forcing
Global model or ERA-Interim reanalysis, with respect to their PDFs. This assessment is performed for the Iberian Peninsula.
Overall, important gains are found for most cases, particularly in precipitation extremes. Most Hindcast models reveal gains above
15%, namely for wintertime, while for precipitation extremes values above 20% are reached for the summer and autumn. As for
the Historical models, although most pairs display gains, regional models forced by 2 GCMs reveal losses, sometimes around -5%
or stronger, for the entire year. However, the spatialization of the DAV is clear in terms of added value for precipitation, particularly
precipitation extremes with significant gains, above 100%.

## 1 Introduction

From the last decades of the 20[th] century up to today, climate change due to anthropogenic gas emissions become a major concern
for mankind. General Circulation Models (GCMs) are the primary tool used by the IPCC (Intergovernmental Panel on Climate
Change) to assess past, present, and future climate conditions. Overall, GCMs can capture the large-scale circulations of the
atmosphere and the ocean, together with their centennial to decadal variability and synoptic weather (Meehl et al., 2007; Randall
et al., 2007; Stocker et al., 2014). However, the GCMs coarse resolution does not allow for a good representation of orography,
land-ocean-atmosphere interactions, or sub-grid processes (Randall et al., 2007; Rummunukainen, 2010; Soares et al., 2012).
Instead, these local processes rely often on parametrizations resulting in a poor description of processes such as convection and
thermal circulations (Prein et al., 2015). Therefore, to bridge the gap between large to local-scale climate, downscaling techniques
were developed. These include statistical downscaling (Wilby et al., 1998; Khan et al., 2006) and dynamical downscaling (Giorgi
and Bates, 1989; McGregor, 1997; Christensen et al., 2007; Rummukainen, 2010), in which the latter makes use of Regional
Climate Models (RCMs). RCMs are run over a geographical domain (continental, national or regional) driven by a GCM, including
reanalysis, by means of the boundary conditions. RCMs are an important tool for the representation of regional to local climates,
since they are run at much higher resolutions, nowadays between tenths of kilometres to the convective permitting scales, and
therefore can capture physically consistent regional to local processes and circulations (Giorgi and Mearns, 1991; 1999; Leung et
al., 2003; Laprise, 2008; Heikkilä et al., 2010; Soares et al., 2012a; 2012b; Cardoso et al., 2013; Rios-Entenza et al., 2014; Soares
et al., 2014). The gains for an individual variable or process of the higher-resolution RCM simulations against lower-resolutions,
given by the GCM or reanalysis driving the RCM, relative to observation, are commonly known as added value (Di Luca et al.,
2012; 2013; Prein et al., 2013a; 2016; Torma et al., 2015; Rummukainen, 2016; Soares and Cardoso, 2018; Cardoso and Soares,
2021; Careto et al 2021; Soares et al., 2021).

In recent years, the increase of computational resources has allowed researchers to run simulations at larger domains and resolutions
(Prein et al., 2015; Soares et al., 2017; Jacob et al., 2020; Coppola et al., 2020). These often encompass an entire continent, spanning
larger periods from a few decades to over a century. For instance, from the 20th to the end of the 21st century or from the 50 km
horizontal resolution from the PRUDENCE project (Christensen and Christensen, 2007) or the 25 Km from the ENSEMBLES
(van der Linden and Mitchell, 2009), down to the 12 Km resolution from the World Research Climate Program Coordinated
Regional Downscaling Experiment (WRCP-CORDEX, Jacob et al., 2014; 2020) for the European domain (EURO-CORDEX,
hereafter). Moreover, other examples are the ideal case studies, employing simulations at kilometre-scale (Hohenegger et al., 2009;
Kendon et al., 2012; 2014; Prein et al., 2013b; Ban et al., 2014; Froidevaux et al., 2014; Fosser et al., 2017; Imamovic et al., 2017;
Leutwyler et al., 2017; Liu et al., 2017; Kirshbaum et al., 2018; Fumière et al.,2020; Berthou et al., 2020), or the convective



permitting simulations from the WRCP-CORDEX Flagship Pilot Studies focused over the Alps (Coppola et al., 2020; Ban et al., 2021; Pichelli et al., 2021).

The evaluation and added value of higher resolution simulations constitute an important step to gauge their quality and usefulness. Soares and Cardoso (2018) proposed a new metric to quantify the added value of higher resolutions with respect to their forcing or lower resolution counterpart simulations. This metric is based on the ability of models in representing the observed probability density functions (PDF). It relies on a distribution added value (DAV) which can be applied to either the full PDF or to PDF sections, thus enabling an easy evaluation of extremes or any section of the PDF.

In the past, the hindcast simulations from the EURO-CORDEX were extensively evaluated, revealing gains for the main meteorological variables (Kotlarski et al., 2014; Casanueva et al., 2016a; 2016b; Prein et al., 2016; Soares and Cardoso, 2018; Herrera et al., 2020; Cardoso and Soares, 2021, Careto et al., 2021). Kotlarski et al. (2014) assessed temperature and precipitation at monthly and seasonal timescales for the hindcast simulations, reporting slight improvements from EURO-CORDEX relative to the ENSEMBLES (Van der Linden and Mitchell, 2009). Overall, the models showed to be able to capture the space-time variability of the European climate. However, when considering averages over large subdomains and at the seasonal timescale, the higher resolution simulations did not reveal noticeable improvements. Prein et al. (2016) also assessed precipitation for both resolutions of the hindcast EURO-CORDEX (50Km and 12Km) and found improvements, mostly in regions characterized by complex terrain and also in summertime precipitation, due to the better resolved convective features. More recently, Herrera et al. (2020) performed an assessment for precipitation and temperature for an ensemble of 8 Hindcast EURO-CORDEX RCMs over the Iberian Peninsula. The authors report a good spatial agreement between models and observations, namely for temperature. On the other hand, this agreement decreases when extremes are considered. Nevertheless, the authors also report a larger uncertainty related to observations for precipitation relative to temperature.

The first to quantify the added value of the EURO-CORDEX hindcast runs were Soares and Cardoso, (2018), evaluating 5 RCMs for precipitation at both resolutions (50Km and 12 Km) considering their probability density functions with station-based dataset as observational benchmark. This study reported relevant added value of the RCMs against the driving ERA-Interim reanalysis (Dee et al., 2011). Nonetheless, when comparing both resolutions, the improvements are not as significant, with the exception of extreme precipitation. More recently, other studies such as Cardoso and Soares, (2021), Soares et al. (2021) and Careto et al, (2021), used the DAVs technique to assess the added value for other variables, simulations and domains.

The precipitation historical period EURO-CODEX simulations were also assessed for specific regions (Torma et al., 2015; Soares et al., 2017; Ciarlo et al., 2020). For instance, Torma et al. (2015) evaluated precipitation over an alpine area, whereas Soares et al. (2017) assessed the same variable but for Portugal. Both studies describe the ability of the higher resolution runs in simulating the mean spatial and temporal patterns of precipitation, as well as their distributions. More recently, Ciarlo et al. (2020) assessed the added value of all available EURO-CORDEX and CORDEX-CORE (Gutowski et al., 2016) simulations for precipitation, also considering a probability density function metric. The authors found added value, particularly at the tail of the distributions, however, they also report a significant uncertainty linked to the observational datasets in the results.

In this study, the DAV metric is used to assess the added value of precipitation for all available 12 Km resolution simulations from the EURO-CORDEX Hindcast (1989-2008) and Historical (1971-2005) set. The added value is then computed comparing the RCMs precipitation results versus their corresponding driver GCM or ERA-Interim reanalysis, where the recently developed Iberia Gridded Dataset (IGD, Herrera et al., 2019) is considered as baseline. The IGD is a high-resolution dataset, with 0.1º resolution, and is based on a large number of weather stations covering the entire Iberian Peninsula. Thus, a new and unprecedent assessment of the added value in the high-resolution EURO-CORDEX regional simulations is performed, with observations at a similar scale.



The next section introduces the data and a description of the methods considered. The results and discussion are presented in the following section. Finally, the main conclusions are drawn in the last section.

## 2 Data and Methods

### 2.1 Iberian Gridded Dataset

A recently developed dataset, the Iberian Gridded Dataset at 0.1º resolution (IGD, Herrera et al 2019) is used as a baseline for the
added value assessment. This dataset was built by considering an unprecedented number of weather stations – 3486 (275) for precipitation (temperatures) -over the entire Iberian Peninsula for daily precipitation, maximum, minimum and mean temperatures, spanning over 45 years, from 1971 until the end of 2015. The authors performed a comparison with E-OBS v17 and v17e, confirming the ability of this new dataset in reproducing the mean and extreme precipitation and also the temperature regimes. Both datasets are comparable, yet statistically different. Since a large number of stations were considered, particularly for
precipitation, IGD should reproduce more realistically the climate of the Iberian Peninsula.

### 2.2 EURO-CORDEX

The aim of CORDEX is to develop a coordinated ensemble of high-resolution Regional Climate projections to provide detailed climate data for all land regions of the world, at user-relevant scales, and support climate change impact and adaptation research (Giorgi et al., 2009; Gutowski et al., 2016). All model data is available at the Earth System Grid Federation portal (Williams et al.,
2011). The EURO-CORDEX (Jacob et al., 2014; 2020) is a branch from the international CORDEX initiative and consists of a multi-model ensemble of simulations at 50Km, 25Km or 12Km resolutions for a European domain. These simulations consist in Hindcast for the 1989-2008 period, forced by the ERA-Interim Reanalysis (Dee et al., 2011), and Historical/Scenario simulations driven by the Intergovernmental Panel on Climate Change Coupled Model Intercomparison Project – Phase 5 (IPCC-CMIP5) GCMs covering the 1971-2100 period. All simulations are available at the Earth System Grid Federation portal (Williams et al.,
2011; https://esgf.llnl.gov/).

The information regarding the simulations used are summarized in Table S1 for Hindcast and Table S2 for the Historical. For all models, the added value is computed for the common Iberian Peninsula domain shown in Fig. 1, where prior to all computations, all RCM model data was first conservatively interpolated (Schulzweida et al., 2009) into this observational domain, while the observations were interpolated into each low-resolution grid. Thus, the evaluation of the EURO-CORDEX regional models is
performed at the 0.1º regular grid, while at the same time the GCMs or ERA-Interim (0.75º) are evaluated at their native resolutions (see table S2 for each GCM resolution).

### 2.3 Distribution Added Value

The Distribution Added Value (DAV) is a metric put forward by Soares and Cardoso (2018) which allows assessing in a direct way the gains or losses of using higher against lower resolution models relying on their probability density functions (PDF), by
having an observational dataset as reference. DAV uses the PDF skill score proposed by Perkins et al. (2007) to measure the similarity between two different PDFs. In order to compute this metric, first, a PDF must be built from the data. In this work, two slightly different methods are considered for building the PDFs to assess the daily precipitation from the EURO-CORDEX models. In the first method is the precipitation values are accumulated within each bin, thus returning a precipitation intensity distribution. While the second one, considers the number of events that fall into each bin, thus returning a precipitation frequency distribution.



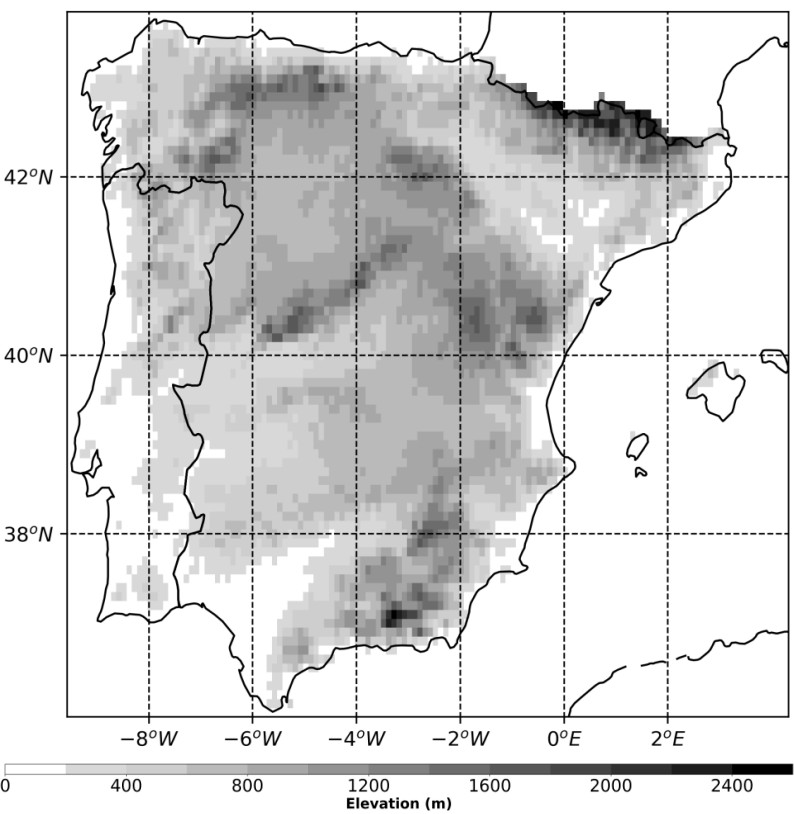

**Figure 1. Orography from the Iberian Gridded Dataset for the Iberian Peninsula at 0.1º horizontal resolution**

Then, a normalization is carried out by dividing each bin by the sum of all bins (Gutowski et al., 2007; Boberg et al., 2009; 2010). With this normalization, one can more accurately compare the results between seasons or regions (Soares and Cardoso, 2018), but also, changes in PDF are identified more straightforwardly (Gutowski et al., 2007). Each bin has a width of 1 $mmday^{-1}$ to avoid

excessively fine and potential noisy steps in both methodologies, thus satisfying the criteria proposed by Wilks, (1995). All DAVs are computed by only considering the wet days, i.e., days with precipitation equal or above to 1 mm, as models tend to overestimate the days with very small precipitation amounts (Boberg et al., 2009; 2010; Soares and Cardoso, 2018). For either methodology, the score is given by the sum of the minimum value obtained at each bin between the models PDF and the observational PDF:

$$S = \sum_{1}^{n} \min\left(Z^m, Z^{obs}\right) \qquad (1)$$

Where n is the number of bins for the PDFs, m denotes the high or the low-resolution simulation and obs is the observational PDF. For precipitation, the limits are bounded between 1 and 300 mm, roughly corresponding to the maximum precipitation rate in IGD. Subsequently, the DAV metric is then computed as follows:

$$DAV = 100 * \frac{S_{hr} - S_{lr}}{S_{lr}} \qquad (2)$$





with the subscript $hr$ depicting the high resolution and $lr$ the low resolution. The DAVs return the fraction or percentage of gains

or losses of value by downscaling the low-resolution runs. With the normalization of the PDFs, the contribution from each bin to the overall score of a particular model is more relevant for the lower bins, decreasing when approaching the tails of the distribution. If for a specific bin there is no model or observation data, then the contribution of that bin would be 0. By definition, the maximum value for S is 1, where if a specific model overestimates the observable PDF in one section, then it will inevitably underestimate in another section. Both these scenarios lower the score of individual models. DAVs is a versatile metric with the advantage of

being able to be computed for PDF sections, which is useful for the extremes added value characterization. In this study, the added value assessment is performed by considering not only the whole PDF but also for an extreme precipitation PDF section, where only values above the observational 95th percentile are accounted for. Since the resolution difference between observations and the high-resolution models is approximately 0.01º, this threshold is computed from the observations at the original resolution, while for the low-resolution driving models, the percentile is obtained from the interpolated observations.

For the DAVs assessment, firstly, a regional approach is considered by pooling together all data from the Iberia Peninsula, thus computing the added value for the entire domain. Secondly, a spatial approach is performed, where all data within each grid cell from the low-resolution simulation is pooled together, returning a DAV's spatial view, instead. Nevertheless, it should be noted that the Iberian overall value does not represent a mean from the spatial DAVs. Although the results should be similar, one must consider that different behaviour are expected, and care must be taken when comparing the results.

## 3 Results and Discussion

### 3.1 Hindcast (1989-2008)

The next subsection presents the results for the EURO-CORDEX Hindcast (1989-2008) simulations, by applying the DAVs metric to precipitation and precipitation extremes. All results have the IGD as a reference. The precipitation PDFs are shown in the supplementary material. Figure S1 root to the results displayed in Fig. 2. Two different approaches are performed, one following

a precipitation intensity PDF (left panels in Fig S1) and a precipitation frequency (right panels in Figures). Overall, the high-resolution RCM simulations capture better the observable PDFs, contrary to their lower-resolution counterparts This behaviour suggests an expected and overall added value of the high-resolution runs relative to. the coarser resolution over the Iberian Peninsula domain, for both the annual and seasonal timescales. Although differences are visible for both methodologies, the precipitation intensity PDF reveals a larger spread between low and high resolution, particularly at the lower bins. Thus, one can

anticipate a generalized stronger added value (positive DAV), due to a closer representation of the RCM PDFs to observations, relative to the low-resolution. Moreover, low-resolution runs tend to overestimate considerably the lower rainfall bins, and in consequence of the normalization, the higher bins, roughly above the 15 mm/day are underestimated. The same occurs for the high-resolution runs, however at a lower degree, hence reproducing more reliably the observable PDF.

In Fig. 2, the DAVs for the entire Iberian Peninsula are shown, revealing important gains of the EURO-CORDEX Hindcast high-

resolution simulations in comparison to the driving ERA-Interim. Fig. 2a displays the DAVs for precipitation intensity considering the whole PDF, with significant gains at the annual scale for 11 models, which have a DAV equal or above to 10%. From these CCLM, ETHZ, CNRM63, and SMHI models stand out surpassing 18%. CNRM53, ICTP, and IPSL RCMs show lower gains, ranging from ~5 to 10%. In winter, spring and autumn the models roughly reproduce similar DAV values seen at the annual scale, however this is not the case for summer. Of all seasons, summer has the lowest performances, particularly CNRM53, DHMZ, and

IPSL showing a detrimental effect ranging from -0.2 % to -6%. In fact, the summer is the season where models display





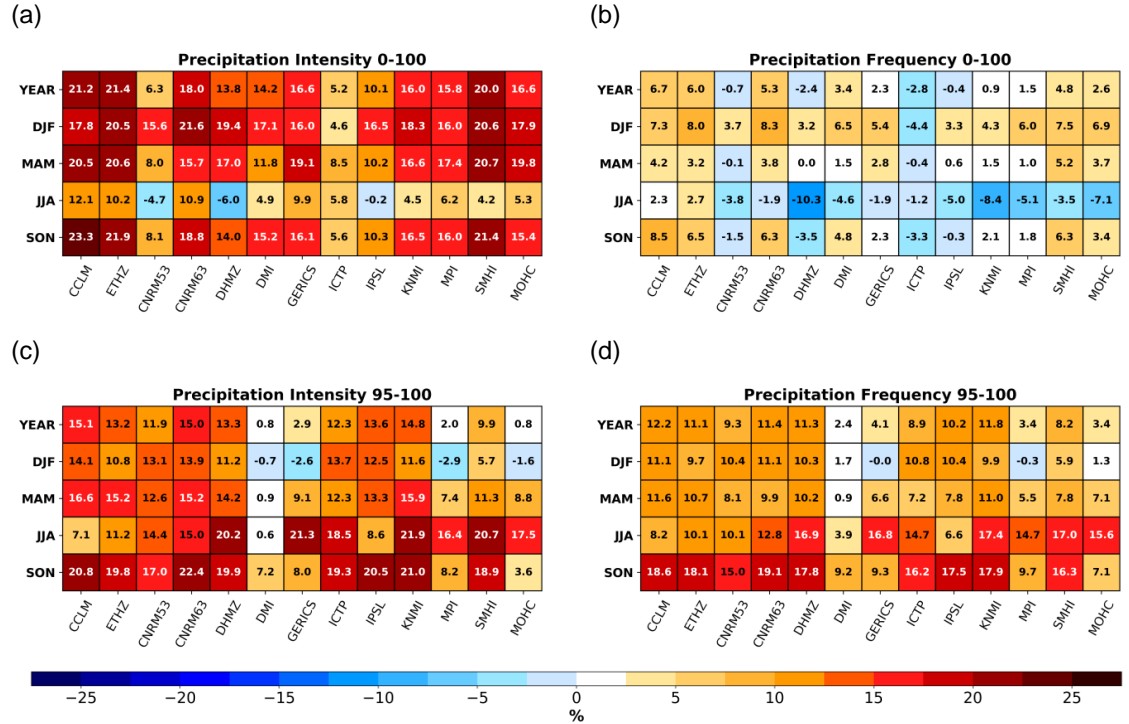

**Figure 2. Yearly and seasonal distribution added values (DAV) of the Iberian Peninsula, between the RCMs and the ERA-Interim reanalysis for the 1989-2008 period, taken from the Hindcast EURO-CORDEX simulations, with the IGD regular dataset as a reference for (a) daily precipitation intensity, considering the whole PDF shown in the left panels of Figure S1, (b) daily precipitation frequency considering the whole PDF shown in the right panels of Figure S1, (c) daily precipitation intensity extremes, only considering the values above the observational 95[th] percentile shown in Figure S1 left side and (d) daily precipitation frequency extremes, only considering the values above the observational 95[th] percentile shown in the right side of Figure S1. All RCM data were previously interpolated to 0.1º regular resolution from the observations, while the observations were interpolated into the ERA-Interim resolution**

more difficulty in capturing the precipitation features, since it is the driest season for the entire Iberian Peninsula, and the lower

precipitation rates are more expressive, which in turn could contribute to uncertainties (Rios-Enteza et al., 2014). In fact, the

summer PDF for the precipitation intensity (Fig. S1), in comparison with the other seasons, reveal a stronger overestimation for

the lowest bins and an underestimation in the tails, thus reducing the downscaling added value.

While Fig. 2a shows the added value for the precipitation intensities, Fig. 2b considers the precipitation frequencies. The overall

DAVs are lower, yet the same models reveal either maximum or minimum DAVs. In this case, 4 models reveal some detrimental

effects associated with the downscaling ERA-Interim at the annual scale. For instance, the underperformance of the downscaling

ICTP is highlighted with the negative values at the yearly scale derived from the poorer performance for winter and autumn.

Similar to the precipitation intensity, summer has the lowest DAV values, but in this case 11 out of 13 models reveal losses, in

particular DHMZ, KNMI, and MOHC with negative DAVs superior to -7%. In fact, Herrera et al., (2020 reported a bad

performance for the DHMZ RCM for all the metrics. In the opposite sense for most RCMs, winter and to some extent in spring

and autumn reveal gains, particularly winter with DAVs above 7% for 4 RCMs. In Herrera et al., (2020), all regional models

assessed against the IGD still reveal strong biases for precipitations, but nevertheless, the gains found here for either precipitation

intensities (Fig. 2a) and frequencies (Fig. 2b) still reveal improvements in comparison to the driving ERA-Interim reanalysis.

Moreover, in Soares and Cardoso, (2018) the Iberia displayed larger added value in comparison to the other regions analysed.



The next panel shows the DAV metric but applied only for values above the observational 95[th] percentile, where the normalization is carried out considering just the bins above the P95, thus only inspecting the added value related to the extreme precipitation tail. Previous studies such as Soares and Cardoso, (2018) and Ciarlo et al. (2020) reported more relevant gains when looking into extremes, with a few exceptions. Instead, results from Fig. 2c reveal lower DAVs compared to the whole PDF case (Fig. 2a), yet still significant. Still in Soares and Cardoso, (2018), for the Iberia Peninsula, despite the low station density considered, the DAVs reveal weaker values for the extremes and stronger for the whole PDF. At the annual scale, 4 RCMs reveal similar performances in comparison to the driving simulation, while the other RCMs have added value above to ~10%. From all seasons, winter reveals 4 RCMs with slight detrimental effects, contrasting with the results obtained from the other seasons and also against Fig. 2a. The downscaling shows a better performance, i.e., a larger added value, for the summer and autumn season for more than half of the RCMs, from which display gains equal or superior to 15%. The gains for the summer season where most precipitation is convective (Azorin-Molina et al., 2014) are relevant as low-resolution models have trouble in capturing these highly spatial and temporal heterogeneities, due to shortcomings associated with the parametrization of convection (Prein et al., 2015). Spring also revealed significant added value for almost all RCMs, ranging from 7.4% to 16.6%. The exception is for the DMI model which has limited to no added value throughout the year, apart from autumn. In fact, this RCM is the only one showing a similar performance to the driving simulation for 3 seasons. In Herrera et al., (2020) most models overestimate the 50-year return period for precipitation extremes for a large part of the domain, which is in line with the results shown in Fig. S1. However, the low-resolution models can't reproduce such high precipitation rates, which in the end results into added value, despite the overestimation.

For the precipitation frequency extremes (Fig. 2d), the DAV values are almost always slightly lower in comparison to precipitation intensity (Fig. 2c), but larger than Fig. 2b. The similarity across both methodologies at the yearly and seasonal scale is clear, where summer and autumn reveal more significant gains, while for spring and winter most RCMs display gains close to 10%. The exceptions are DMI, GERICS, MPI, and MOHC, namely for winter and autumn with more limited gains.

Figure S2 from the supplemental material displays a slightly different approach than in Fig. 2. In this case, all model data was previously interpolated to the 0.1° resolution from the observations. A conservative remapping of precipitation was considered (Schuzweida et al., 2009), resulting in a smoothing precipitation field, which do not significantly impact the PDFs from ERA-Interim. On the other hand, by upscaling the IGD, part of the original variability is conserved, thus changing the intensity or frequency within each bin. These differences result in stronger gains for Fig. S1 in comparison to Fig. 2. Nevertheless, the overall inter-model variability is close between these two metrics.

Figure 3 displays the same metric used in Fig. 2, but from a spatial overview. Here a different approach is implemented, where the percentiles and PDFs are computed by pooling together all information only within each grid cell from the low-resolution driving model. Fig. 3a displays a spatialization of the added value for the full precipitation intensity PDF, revealing a significant added value for most of the domain, as expected from the results obtained from Fig. 2a. In general, the gains are larger for the coastal areas and the Pyrenees throughout the year, due to a better representation by the higher-resolutions of the land-sea and topographically induced circulations. The more expressive gains for coastal areas, particularly for the Mediterranean, were also observed in Careto et al 2021 for maximum and minimum temperatures. Similar to Fig. 2a, most models for winter, spring, and autumn mostly display locations with significant added value. For summer, the gains are more focused in the southern part of the peninsula, albeit most models still reveal significant percentages for the central and northern locations. RCMs such as CNRM53, ICTP, and to some extent IPSL reveal sites in the Iberian interior with low or even negative added values. Those values have a







**Figure 3. Yearly and seasonal spatial distribution added values (DAV) of the Iberian Peninsula, between the RCMs and the ERA-Interim reanalysis for the 1989-2008 period, taken from the Hindcast EURO-CORDEX simulations, with the IGD regular dataset as a reference, for (a) daily precipitation intensity, considering the whole PDF, (b) daily precipitation frequency considering the whole PDF, (c) daily precipitation intensity extremes, only considering the values above the observational 95th percentile from each ERA-Interim grid point and (d) daily precipitation frequency extremes, only considering the values above the observational 95th percentile from each ERA-Interim grid point. All RCM data were previously interpolated to 0.1º regular resolution from the observations, while the observations were interpolated into the ERA-Interim resolution.**

clear impact on the results shown in Fig. 2a. Moreover, DHMZ which was revealed to have the minimum value in the summer regional overview, displays some points in Fig. 3a with having a slightly detrimental effect, ~-10%.

In contrast with Fig. 3a and following the results from relative precipitation frequency in Fig. 2b, Fig. 3b shows an overall smaller added value, following a very similar inter-model difference. While for precipitation intensity, values easily surpassed 50%, namely near the coast, here the gains are more limited, going up to 30% in some coastal sites. Moreover, for the same models which had a lower performance in the regional overview, particularly over the summer season, locations emerge having DAVs of ~-10 % to -15 % in comparison with the driving simulation. Nevertheless, for the same season, all models reveal significant added value for the southern Peninsula, mirroring Fig. 3a. However, these positive values are not enough to reverse the losses found for the entire domain in Fig. 2b. The regional overview of the DAVs is not a mean from the spatializations and thus, care must be taken when comparing both figures. For instance, the ICTP model in Fig. 2b had the worst overall performance with negative values for all time scales, while in Fig. 3b the picture is different. ICTP only shows negative percentages at the annual and for the winter and autumn seasons. On the other hand, DHMZ which had the minimum DAV in Fig. 2b, still reveals locations with significant gains.

The picture for the extreme precipitation intensity and frequencies is completely different (Figs. 3c and 3d). Here the gains of the high-resolution relative to the ERA-Interim reanalysis are evident when moving to more local scales. Moreover, the contrast of values with the extremes shown in Fig. 2, highlights the difference in both methodologies. For the spatialization, each point of the low resolution only considers the amount of information available within, whereas, in Fig. 2, all data is considered. In the former, there are fewer values above the observational percentiles and even less for the seasons, contrasting with the high-resolution. This fact highlights the difficulty in the representation of extremes by the lower resolutions models, resulting in added value. Moreover, in Fig. 2, precipitation intensity showed stronger DAVs, while for Figs. 3c and 3d, the results are similar with points revealing gains above 250%. Still, at the annual scale, in comparison to the individual seasons, it is possible to see lower percentages over the centre of Iberia, highlighting the improvements near the coast.

### 3.2 Historical (1971-2005)

The next section displays the same metric, but applied to the Historical simulation, covering the 1971-2005 period. For this case, the same RCM could be forced by different GCMs, however, the results do not necessarily have to agree. In fact, following the values from Fig. 4, the different performances are more closely related to the different GCMs themselves than across the same high-resolution models. This enforces a weak or even no relationship between a single RCM forced by different GCMs. Moreover, any comparison with the previous Hindcast (Figs. 2 and 3) is hindered, not only owing to these differences, but also due to different time periods. Nevertheless, the range of DAVs for the Historical is considerably higher. Figs. S3 and S4 display the PDFs of models and observations. In the first all model data was previously interpolated to the 0.1º resolution of the IGD observations, as for the last, the PDFs from the GCMs are kept at their original resolutions, while the observations were interpolated into each GCM grid. Thus, the PDFs for the RCMs in Fig. S3, together with the PDFs in Fig. S4 root to the DAVs shown in Fig. 4. Following the





results from the Hindcast simulations, there is a stronger agreement between all PDFs for precipitation frequency (right side of Figs. S3 and S4) in comparison to precipitation intensity (left side of Figs. S3 and S4), namely for the lower bins, anticipating a more difficulty in obtaining added value for the last.

The normalized precipitation with respect to the whole PDF is shown in Fig. 4a. For most cases, there exists some or even a significant added value. From those, models forced by the NCC GCMs are highlighted with having the highest added values for all time scales, namely at the annual scale, with values above the 30%. For this GCM group, only the winter season displays slightly lower gains for some RCMs, yet still significant. IPSL-LR-GERICs also display significant gains at the annual scale, reaching 28%, contrasting with the slightly weaker results for the individual seasons. The other GCM groups do not reveal such a relevant

added value, namely models forced by CNRM, ICHEC1, ICHEC2, IPSL-MR, NOAA with gains ranging between 5% to 20%. Despite the overall gains, some models driven by CNRM for winter, spring, and summer reveal an absence of added value. Moreover, all ICHEC1 driven RCMs for winter, ICHEC1-DMI for spring, and some RCMs driven by ICHEC2, particularly for summer, reveal weak and sometimes slightly negative percentages bounded between -5.7% to 0.4%. While the previous pairs still reveal some gains, MPI1 and MPI3 mostly show neutral percentages, indicating a closer performance between low and high

resolutions. As for MPI2, the RCMs driven by this GCM reveal negative percentages for all time scales, having the worst results amongst all GCM-RCM pairs. Following the previous results, the MOHC driven RCMs also show weak gains and even some detrimental effects in 4 RCMs at the annual scale. From these same RCMs, at least one season also displays losses up to -6.9%. Nevertheless, MOHC-ETH and MOHC-KNMI are still able to reveal some added value.

The different behaviour across the downscaling of each GCM group may be related not only to their resolution but also to the

performance and quality of the GCMs itself, mainly within the lateral boundary forcing zone. For instance, Brands et al. (2013) describes the MPI and MOHC as the best GCMs over these regions. On the other hand, Jury et al. (2015) refers to the IPSL_MR as having a poor performance for upper-air variables over the same forcing region. Moreover, table 6 from McSweeney et al. (2015) resumes the overall performance for the individual GCMs, with CNRM, NOAA, MOHC and MPI as having a good performance; IPSL_MR and NCC with an intermediate performance; ICHEC and IPSL_LR with a poorer performance. In fact,

the RCMs driven by either the MOHC or MPI GCMs have a difficulty in obtaining added value, whereas the models driven by NCC or IPSL_LR clearly display added value. The resolution of the GCMs can also play a major role in the added value of precipitation. Although NOAA GCM reveals a good performance (McSweeney et al., 2015), at the same displays one of the lowest resolutions, which may be a possible reasoning behind the gains found in Fig. 4a for the NOAA-GERICS pair.

The next panel displays the precipitation frequency relatively to the whole PDF (Fig. 4b). As with the Hindcast simulation (Fig.

2), the frequency reveals limited gains, but with similar inter-model differences, correlating well with the precipitation intensity approach. Thus, the overall results for precipitation frequency are weaker and closer to 0%. In other words, the negative values are not as strong, where for instance the losses for the MPI2 driven RCMs are slightly less significant in comparison to Fig. 4a. At the other end of the spectrum, models forced by the NCC GCM still reveal a more significant added value, albeit weaker when compared to precipitation intensity. The exception is for winter where 6 RCMs display weaker and slightly negative DAVs.

Following this reasoning, the other GCM-RCM pairs reveal a similar inter-model variability across Figs. 4a and 4b. The main reason for the similar behaviour, yet weaker results, come from the closer PDFs seen for precipitation frequency in comparison to precipitation intensity (Figs. S3 and S4).

Figure 4c displays the results for normalized precipitation extremes by considering only the values above the observational 95[th] percentile. Comparing with the Hindcast results from Fig. 2, the DAVs, in this case, do display some GCM-RCM pairs with






**Figure 4.** Yearly and seasonal distribution added values (DAV) of the Iberian Peninsula, between the RCMs and the CMIP5 GCMs for the 1989-2008 period, taken from the Historical EURO-CORDEX simulations, with the IGD regular dataset as a reference for (a) daily precipitation intensity, considering the whole PDF shown in the left panels of Figure S3, (b) daily precipitation frequency considering the whole PDF shown in the right panels of Figure S3, (c) daily precipitation intensity extremes, only considering the values above the observational 95[th] percentile shown in Figure S3 left side and (d) daily precipitation frequency extremes, only considering the values above the observational 95[th] percentile shown in the right side of Figure S1. All RCM data were previously interpolated to 0.1º regular resolution from the observations, while the observations were interpolated into each CMI5 GCM resolution.





stronger added values, thus showing higher variability. Mirroring Fig. 4a, the NCC forced RCMs reveals a very significant added
value at the annual scale, derived from the strong signal found seasonally, particularly for winter. At the same time, 4 RCMs from
this group are highlighted for having gains superior to 30% for almost all seasons. The other GCM-RCM pairs do not reveal such
expressive added value. Nevertheless, 2 RCMs driven by ICHEC1, 6 by ICHEC2, and all models forced by the IPSL GCMs,
display at least one season with percentages above 20%. From these, 3 GCM-RCM pairs reveal gains superior to 30% for a single
season. On the contrary, the DMI RCM forced by both ICHEC reveals weaker gains, namely for spring and summer. Moreover,
the IPSL-MR RCMs display weak percentages for spring and even losses for winter, which overshadows the gains found in summer
and autumn. The CNRM driven RCMs reveals 5 models with similar percentages found in Fig. 4a. However, CNRM-DMI and
CNRM-GERICS display losses at the annual scale derived from a stronger detrimental effect for summer and winter respectively.
Not always the extremes reveal added value in comparison to the whole PDF. The losses found in extremes hint towards a lower
accuracy for the RCMs in representing the higher bins. On the other hand, the models forced by MPI1 show an overall
intensification of either gains or losses found in Fig. 4a, meaning that the gains for the whole PDF are possibly derived from the
gains obtained for the extremes. The same occurs for the NOAA-GERICS pair. As for models driven by MPI2 or MPI3, 5 RCMs
reveal gains at either the annual scale or winter, while MPI2-MPI shows losses throughout the year. For these models, although
losses were reported in Fig. 4a, some cases reveal a better representation of extremes. Finally, 6 RCMs driven by MOHC reveal
added value at the annual and seasonal scale, namely for the summer season. On the contrary, MOHC-ICTP displays losses
throughout the year, following the results obtained in Fig. 4a. Moreover, MOHC-CCLM (MOHC-MOHC particularly for summer
(winter) displays some detrimental effects, contrasting with the significant added value for winter (summer). The last panel from
Fig. 4 shows the precipitation frequency. In this case, the similarity across precipitation intensity and precipitation frequency are
evident as there is a good agreement between Figs. 4c and 4d, although as expected from before, with weaker DAVs.

Similar to the Hindcast simulations, a second metric was implemented, where all data were interpolated to the 0.1° resolution from
the observations (Fig. S5). The results here reveal an overall stronger added value due to the stronger effect in upscaling the
observations in comparison to the downscaling of the low-resolution. The gains are even more evident for both precipitation
intensity and frequency extremes, with percentages well above 100% for models forced by the NCC GCM at the annual and winter
season. Nevertheless, the DAVs in Fig. S5 correlate well with the results obtained in Fig. 4.

The next set of figures shows a spatialization of the DAV metric, where individual PDFs and percentiles thresholds were considered
for each low-resolution grid point. As with the Hindcast in Fig.3, the spatialization of the technique allows the emergence of points
with either significant added value or losses, which would be masked otherwise. Figure 5 displays the results for precipitation
intensity, and overall, important gains are found for all models, even for those which underperformed at the Iberia Peninsula scale
in Fig. 4a. Nevertheless, it is possible to verify a more significant added value in coastal areas relative to inland points. This
situation also occurred for the Hindcast simulations and is owed mainly to a better representation of the land-sea circulations.
Moreover, RCMs forced by IPSL-LR, NCC, and NOAA, reveal most grid points with significant gains, corroborating the results
from Fig. 4a. On the contrary, GCMs that displayed gains not as relevant, such as CNRM, ICHEC1, ICHEC2, or IPSL-MR, all
display points with limited gains and sometimes small losses for sites in the interior, thus lowering the joint performance.
Nevertheless, these pairs reveal substantial added value, in particular on the Mediterranean coast. Similar to the previous cases,
models forced by all three MPI GCM versions reveal a similar behaviour, with significant gains in coastal areas and lower values
in the interior. However, these results contrast with the DAVs found in Fig. 4a, namely for models forced by MPI2, i.e., when
assessing the precipitation at a more local scale, the gains become even more evident. Lastly, models forced by

**Normalized Precipitation 0-100**

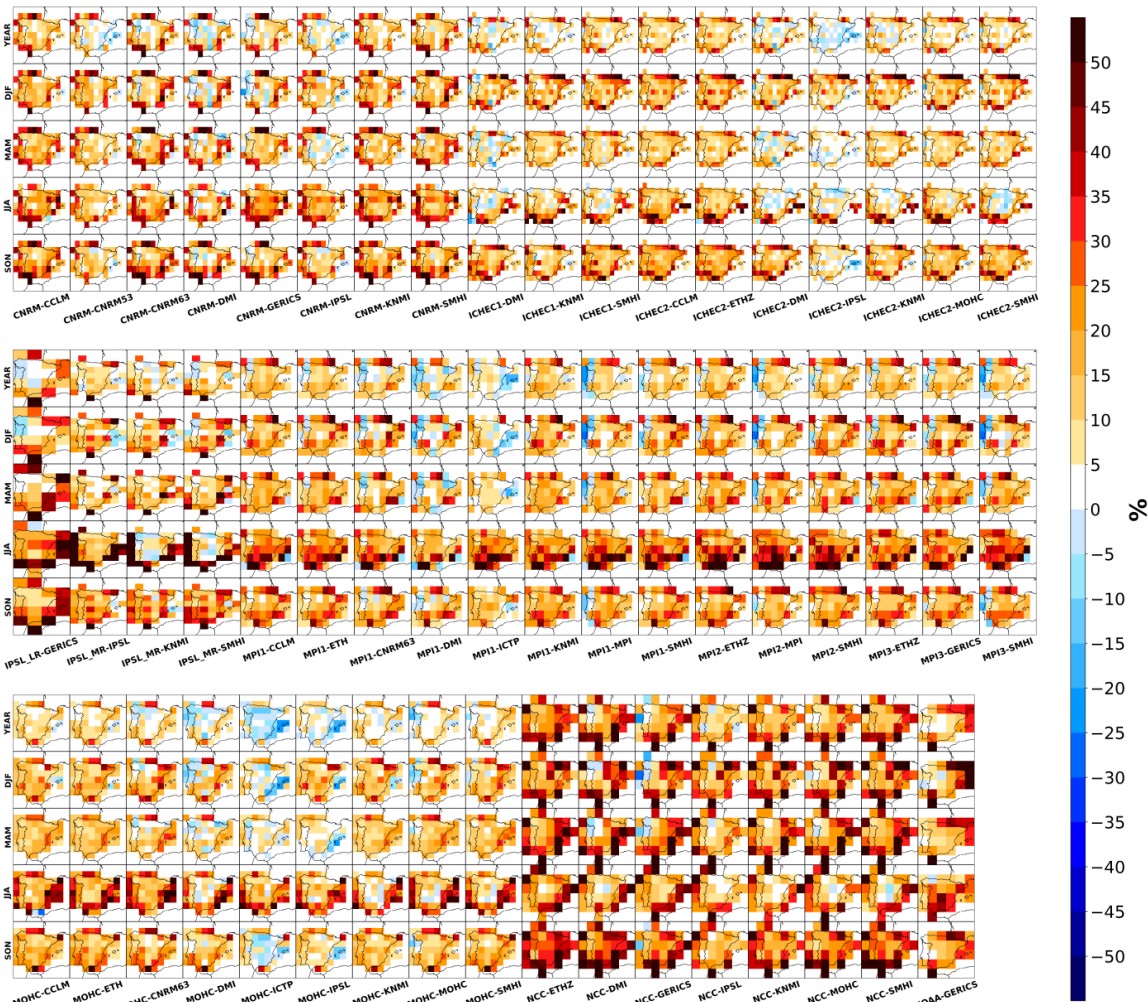

**Figure 5. Yearly and seasonal distribution added values (DAV) of the Iberian Peninsula for the Historical (1971-2005) EURO-CORDEX RCMs, with the IGD as a reference for the daily precipitation intensity, considering the whole PDF. All RCM data were previously interpolated to 0.1º regular resolution from the observations, while the observations were interpolated into each CMI5 GCM resolution.**


the MOHC GCM still reveal relevance for the most part, namely for summer in coastal areas, contrasting with the weaker values in the other seasons. MOHC-ICTP had the overall worst performance, yet some added value is still shown for points located on the Atlantic coast for winter, spring, and autumn.

Fig. 6 displays the same results, but for precipitation frequency instead. As expected from the previous cases, the overall gains are

more limited for both the positive and negative percentages. Still, a good agreement with Figs. 4b and 5 are found. From all GCM-RCM pairs, 36 models reveal a better performance for most points within the domain for the summer season. Although a good performance is revealed for most locations for the ICHEC driven GCMs in Figure 5, a widespread weak DAVs occurs for precipitation frequency. IPSL driven RCMs also show relevant gains, at the annual and seasonal scales, namely over Portugal and

**Precipitation Frequency 0-100**



**Figure 6. Yearly and seasonal distribution added values (DAV) of the Iberian Peninsula for the Historical (1971-2005) EURO-CORDEX RCMs, with the IGD as a reference for the daily precipitation frequency, considering the whole PDF. All RCM data were previously interpolated to 0.1º regular resolution from the observations, while the observations were interpolated into each CMI5 GCM resolution.**

on the Atlantic coast. As for the models forced by the NCC GCM, precipitation frequency revealed weaker DAVs in comparison to Fig. 5. Only a single point located over Gibraltar displays a significant added value, above 30% for all seasons but summer.

The next figures show the results for precipitation intensity extremes (Fig. 7) and precipitation frequency extremes (Fig. 8). In comparison to the Hindcast simulations, the results here reveal more spatial variability. Nevertheless, the added value is more significant for the DAVs spatialization, rather than for the whole domain (Fig. 4), particularly for precipitation extremes due to the fact this variable is highly localized. In contrast between Figs. 5 and 6, where some differences arose, the precipitation intensity and frequency extremes reveal a close behaviour. Similar to the previous cases, since the data is further split up when assessing the seasons, the DAVs at the annual scale are usually slightly weaker. In fact, all models in Figs. 7 and 8 clearly reveal

**Normalized Precipitation 95-100**

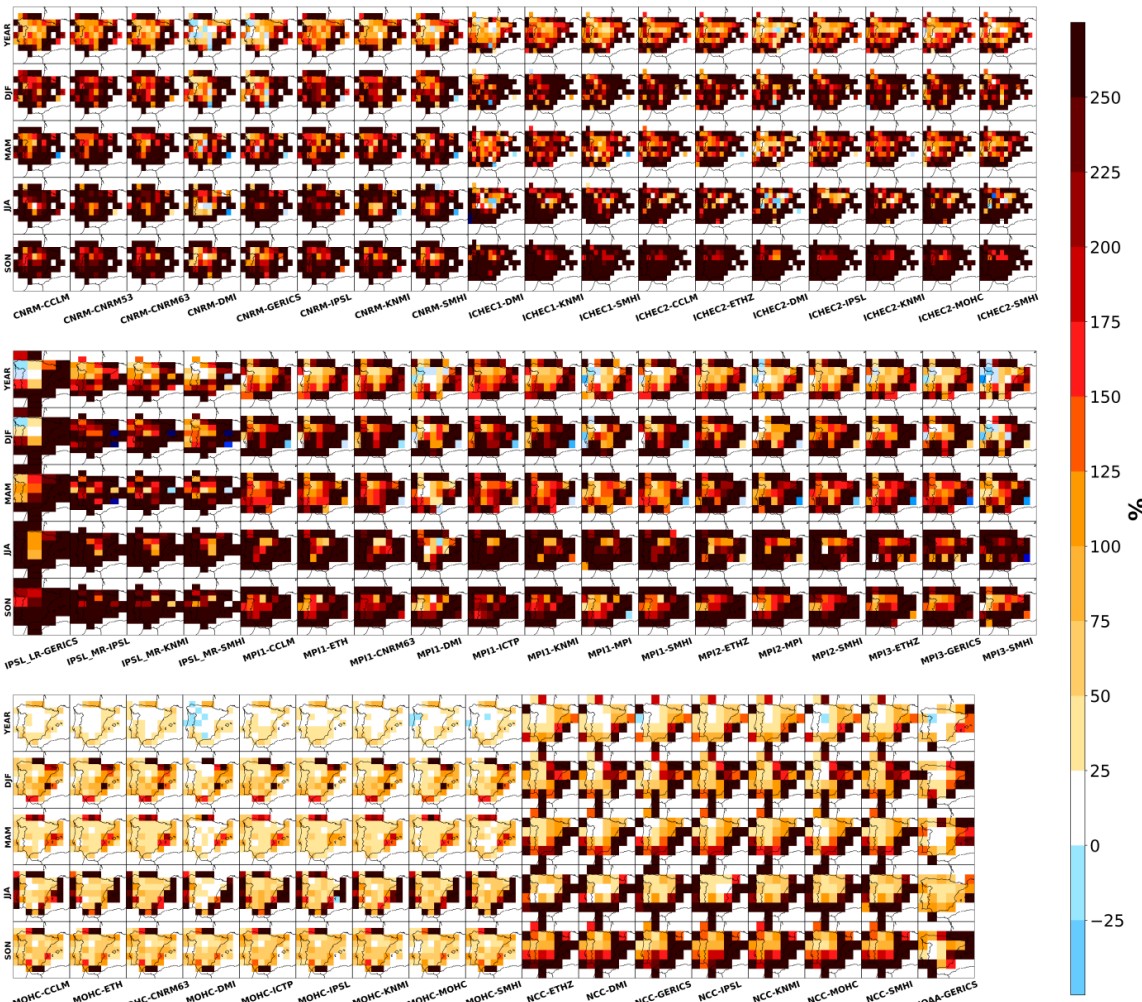

**Figure 7. Yearly and seasonal distribution added values (DAV) of the Iberian Peninsula for the Historical (1971-2005) EURO-CORDEX RCMs, with the IGD as a reference for the daily precipitation intensity, only considering the values above the observational 95th percentile. All RCM data were previously interpolated to 0.1º regular resolution from the observations, while the observations were interpolated into each CMI5 GCM resolution.**


this nature. Moreover, from all model pairs, only the MOHC, NCC, and NOAA groups display weaker DAVs for points in the interior, while the other pairs mostly show significant gains throughout the entire domain.

**4 Discussion and conclusions**

In this study, the performance of RCMs from the Hindcast (1989-2008) and Historical (1971-2005) simulations is assessed

relatively to their PDFs, by using a distribution added value metric proposed by Soares and Cardoso, (2018). This assessment has the IGD regular gridded dataset observations as a reference, covering the entire Iberian Peninsula. To this end, all RCM information

**Precipitation Frequency 95-100**

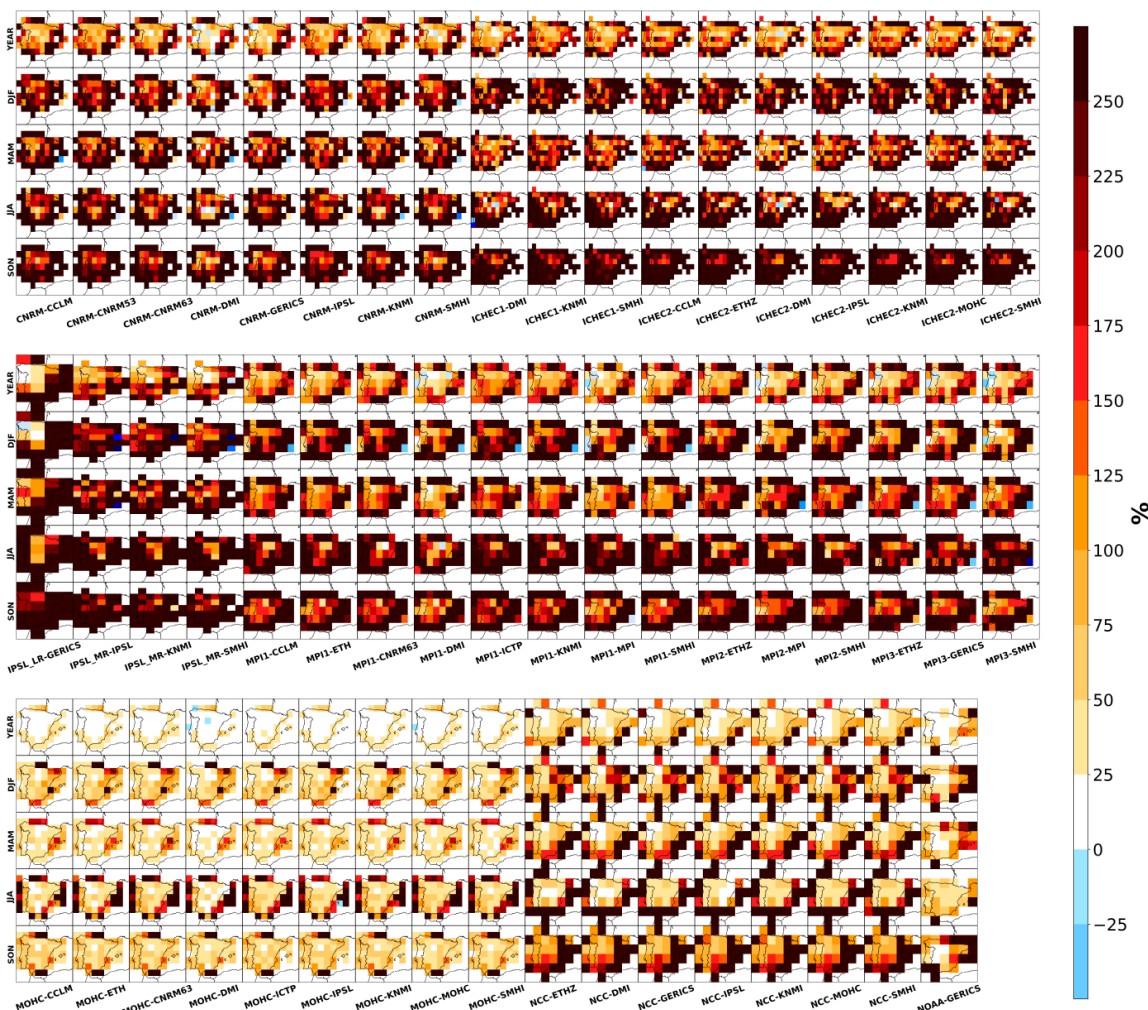

**Figure 8. Yearly and seasonal distribution added values (DAV) of the Iberian Peninsula for the Historical (1971-2005) EURO-CORDEX RCMs, with the IGD as a reference for the daily precipitation frequency, only considering the values above the observational 95th percentile. All RCM data were previously interpolated to 0.1º regular resolution from the observations, while the observations were interpolated into each CMI5 GCM resolution.**


was first interpolated to the 0.1º resolution from the observations, while the low-resolution is assessed at their native resolution. Two slightly different approaches were considered here, one following a precipitation intensity PDF and the other a precipitation frequency. Between both, the results reveal very similar inter-model differences, however, a stronger signal is found for

precipitation intensity. Nevertheless, all RCMs reveal a significant added value, particularly in the representation of extremes, where the global models have more difficulty in describing the higher precipitation rates. This result is expected and shows the importance of considering regional models with higher resolution. However, in some isolated cases, the RCMs instead display neutral or even a slight deterioration effect. On the other hand, the spatialization of the DAVs, in particular for extreme precipitation



revealed significant added values within the entire Iberia Peninsula. These gains are more relevant for coastal sites possibly owed

to the better representation of the land-sea boundary.

Previous works warned about the uncertainty owed to interpolation procedures (Ciarlo et al., 2020). In a way interpolating the GCMs to higher resolutions could generate unrealistic values, whereas upscaling the high resolution degrades the spatial information, affecting primarily the tail end of the distributions (Torma et al., 2015; Prein et al., 2016). To gauge these differences a second methodology was investigated, where all data were interpolated to the 0.1º resolution from the observations. In this case,

the overall DAVs revealed more significant added value. These results hint towards a stronger effect in the upscaling of the high-resolution PDF, approaching ERA-Interim and the CMIP5 GCMs, against the effect of the generation of spurious values when interpolating lower-resolution datasets. Nonetheless, since unrealistic values may be created, the uncertainty associated is higher. While the DAVs metric allows for quantification of the gains or losses by the downscaling of the global models, no relationship is found when the same RCM is forced by multiple GCMs. Yet, a strong connection is observed for high-resolution models driven

by the same GCM. A reasoning for these differences could be primarily attributed to the individual resolution of each GCM, but also to the performance of the GCM along the regions of lateral forcing for the EURO-CORDEX. In this sense, lower-resolution models will show higher DAVs values, although other effects could also play a relevant role, such as model configuration or the parameterizations used. However, if a specific GCM reveals a good performance, then the regional models will have difficulty in obtaining added value. Nevertheless, the gains obtained from the use of higher resolution RCMs are paramount, not only owed to

finer details in the representation of variables by itself, but also due to the increased description of orography, and land-ocean-atmosphere feedbacks, which all have important impacts on precipitation.

**Data availability**

All model and observational datasets are publicly available. The regional and global model data is available through the Earth System Grid Federation portal (Williams et al., 2011; https://esgf.llnl.gov/). The ERA-Interim reanalysis is available at the

ECMWF portal (https://www.ecmwf.int/). The Iberia01 dataset is publicly available through the DIGITAL.CSIC open science service (Herrera et al., 2019a, https://doi.org/10.20350/digitalCSIC/8641).

**Author contribution**

J Careto computed all results and developed the manuscript with contributions from all co-authors. PMM Soares and RM. Cardoso developed the metric in which this paper is based on. S Herrera, J Guitérrez, PMM Soares and R Cardoso previously developed

the observation based Iberian Gridded Dataset.

**Competing interests**

The authors declare that they have no conflict of interest.





**Acknowledgments**

The authors would like to thank all the individual participating institutes, listed in Tables S1 and S2, and the Earth System Grid
Federation infrastructure for providing all the model data used in this study. The authors also acknowledge the Iberian Gridded
dataset (IGD) http://hdl.handle.net/10261/183071.

**Financial support**

J Careto is supported by the Portuguese Foundation for Science and Technology (FCT) with the Doctoral Grant
SFRH/BD/139227/2018 financed by national funds from the MCTES, within the Faculty of Sciences, University of Lisbon. PMM
Soares and RM Cardoso are supported by the FCT under the project LEADING (PTDC/CTA-MET/28914/2017). This work was
also supported by project FCT UIDB/50019/2020 - Instituto Dom Luiz (IDL).

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
