# Peer review of "Added value of the EURO-CORDEX high-resolution downscaling over the Iberian Peninsula revisited. Part I: Precipitation"

_Geoscientific Model Development, 2021_

## Author Response (AR1)

Anonymous Referee #1 on "Added value of the EURO-CORDEX high-resolution downscaling over the Iberian Peninsula revisited. Part I: Precipitation"

General comments to Referee #1

We are very grateful for your kind and positive comments and suggestions. We appreciate all of them. We sincerely think that your revision allowed an overall improvement of the manuscript

**RC1:** This study presents a comprehensive assessment about the added value of precipitation dynamically downscaled regional climate model (RCM) simulations from EURO-CORDEX initiative. To quantify and spatially characterize RCMs performance compared to the corresponding lower-resolution global scale driving fields, Authors take advantage of a distribution-based metric (DAV) previously introduced and presented in Soares and Cardoso (2018). The evaluation regards all the available ERA-Interim reanalysis and global climate models (GCM) driven RCM simulations corresponding to the Hindcast (1989-2009) and Historical (1971-2005) experiments respectively. All the simulations considered refer to the Iberian Peninsula domain and an observational-based Iberian Gridded Dataset (IGD). The present research involves a relevant research question namely if and eventually at what extent downscaled simulations can improve the large-scale forcing signal. This represents a very important point as RCMs are extensively used by a broad range of end users belonging to climate impacts and climate services communities. The main value of the study is to consider the largest dataset of RCMs available and to consider a simple and straightforward metric identifying RCMs potential added value over the entire statistical distribution.

It follows some general, minor remarks:

**RC1:** -Please better clarify what are the main differences in DAV configuration and application respect to the originally work authored by Soares and Cardoso 2018. Do they consist on considering a larger evaluation and historical period simulations and diving DAV according to precipitation intensity and frequency distribution?

**AC:** The main differences between this work and the original Soares and Cardoso (2018) are in the number of simulations for the Hindcast runs, the reference observational database, and the addition of the analysis of the Historical simulations. Here we used all the available 11° resolution EURO-CORDEX 11° simulations, while Soares and Cardoso (2018) considered only the Hindcast which had in common the 0.44° and 0.11° resolutions. Beyond the differences in the models, for this work we used a different observational dataset. In Soares and Cardoso (2018), the authors have the ECAD station data as reference, which does not properly cover the Iberian Peninsula and here we consider a high-resolution dataset, at 0.1°, the Iberian Gridded Dataset (Herrera et al., 2019), which has a very close resolution to that of the EURO-CORDEX models. This new dataset considers more than 3000 stations for precipitation within the Iberian Peninsula, while the ECAD only has a few stations in Spain and just one for Portugal.

Additionally, the methodology splits precipitation into intensity and frequency, while in Soares and Cardoso (2018) only intensity is scrutinised. Here, a spatial distribution of the added value is also provided, while this analysis is absent from Soares and Cardoso (2018).

In the manuscript we have the following:

Line 92: "The first to quantify the added value of the EURO-CORDEX hindcast runs were Soares and Cardoso (2018), evaluating 5 RCMs for precipitation at both resolutions (50 km and 12 km) considering their probability density functions with the station-based dataset ECAD (Klein Tank et al 2002, Klok & Klein Tank 2009) as observational benchmark. This study reported relevant added value of the RCMs against the driving ERA-Interim reanalysis (Dee et al., 2011). Nonetheless, when comparing both resolutions, the improvements are not as significant, with the exception for extreme precipitation."

Line 230: "The same was reported by Soares and Cardoso (2018) for the Iberia Peninsula, despite the low station density considered, the DAVs reveal smaller values for the extremes and higher for the PDF as a whole."

**RC1:** -I think that an important point of the article is the to some extent poor RCMs performance in reproducing summer precipitation intensity and especially frequency (when the entire statistical distribution is considered). Since summer season generally presents a weaker forcing large scale signal it is relevant that the higher resolution self-generated signal (from RCMs) frequently leads to detrimental effects. I think that this aspect deserves some more discussion. It is interesting also considering that this happens mainly when RCMs are driven by ERA- Interim. Finally, this aspect can have also potential relevant propagating effect on the summer temperature representation.

**AC:** We thank the reviewer for pointing out the issue, however the disentangling of the root causes of the lower DAV in summer are out of the scope of the article. Nevertheless, we rewrote the paragraph from lines 195 to 207, to aide the interpretation of the lower added value.

" In fact, summer is the season where models display more difficulty in capturing the precipitation features, since it is the driest season for the entire Iberian Peninsula and precipitation is mostly associated to water recycling through soil moisture atmosphere feedbacks (Rios-Entenza et al., 2014). In addition to the added importance of lower precipitation rates which models overestimate (Boberg et al. 2009; 2010; Soares and Cardoso, 2018), the representation of soil moisture in any model is still very challenging thus the weaker performance of the RCMs is not surprising. In fact, the summer PDF for the precipitation intensity (Fig. S1), in comparison with the other seasons, reveal a higher stronger overestimation for the lowest bins and an underestimation in the tails, thus reducing the downscaling added value. Additionally, ERA-interim assimilates soil moisture and temperature, near surface temperature and humidity thus constraining the local land-atmosphere feedbacks and improving its added value.".

It follows some line-specific, minor remarks:

**RC1:** Line 88. It is not clear the meaning of the "namely for temperature".

**AC:** We thank the reviewer for pointing out the issue and thus decided to remove this part

**RC1:** Line 147. Is the normalization performed for both intensity and frequency distributions?

**AC:** Yes, so that the sum of each individual PDF is equal to 1.

**RC1:** Please be better specify what you mean with the statement: "sum of the all bins".

**AC:** In order to build an empirical Probability Density Function (PDF) from the data we have to bin the data since a theoretical PDF is not considered. However, and particularly for the precipitation intensity, the PDFs between models and observations can reveal some differences. Thus, the normalization of the PDFs. In order to do that we divide each bin by the sum of all bins, or in other words, by the sum of all data considered to build the PDF. To avoid confusion, we decided to change "sum of all bins" to "sum of all data considered as input for the PDF".

**RC1:** Lines 172-173. Please better explain the statement: "Nevertheless, it should be noted that the Iberian overall value does not represent a mean from the spatial DAVs"

**AC:** What we meant here is explained in the prior sentences. The regional value in Figures 2 and 4 does not represent a mean from the spatial values in Figures 3 and 5 to 8. The first results from pooling together all data, and the later results from only pooling together the information within each low-resolution grid cell in order to compute the DAVs. To clarify the connection, we changed "Nevertheless, it should be noted that…" to "Therefore …"

**RC1:** Line 181. An end-phrase dot is missing. Whereas at line 182 there is a misplaced dot.

**AC:** Corrected.

**RC1:** Lines 183-186. These two statements are not clear to me.

**AC:** We thank the reviewer and changed these phrases to: "Contrasts are visible for both methodologies, where for precipitation intensity, the differences between the low- and high-resolution PDFs are more perceptible, particularly at bins below the percentile thresholds. Thus, one can anticipate a generalised larger added value. On the other end, for precipitation frequency, and for the lower bins, the pdfs show a closer representation, almost overlapping, resulting in lower DAVs."

**RC1**: Line 205. What do you mean with "expressive"?

**AC:** We thank the reviewer for the suggestion and changed "… are more expressive…" to "…have an added importance…".

**RC1:** Line 209. "yet the same models reveal either maximum or minimum DAVs." It is not clear, please rephrase.

**AC:** Corrected. We change the statement "The overall DAVs are lower, yet the same models reveal either maximum or minimum DAVs." to "The overall DAVs are lower, yet the models reveal similar differences, with the same models showing maximum DAVs in Figure 2a also present in Figure 2b. ".

**RC1**: Line 289. "the results do not necessarily have to agree." If we consider the same RCM driven by reanalysis.

**AC:** Corrected.

**RC1:** Caption Fgiure4. Please specify the tick blues line RCMs clustering as function of the different driving GCM.

**AC:** We thank the reviewer for the suggestion and added the following statement: "The thick blue lines separate the RCMs driven by different GCM."

Anonymous Referee #2 on "Added value of the EURO-CORDEX high-resolution downscaling over the Iberian Peninsula revisited. Part I: Precipitation"

General comments to Referee #2

**AC:** We are very grateful for your kind and positive comments and suggestions. We appreciate all of them. We sincerely think that your revision allowed an overall improvement of the manuscript

**RC2:** The manuscript assesses the added value of EURO-CORDEX simulations for precipitation over the Iberian Peninsula. The metric "distribution added value" (DAV) is applied for this purpose. The EURO-CORDEX simulations are dynamical downscalings with Regional Climate Models (RCM) of either simulations with a general circulation model (GCM) or a global reanalysis. The DAV is then computed as the percentage change in the Perkins skill score for probability density functions derived from the RCM simulations and their respective lower-resolution driving datasets. As an observational reference for this evaluation, the Iberian Gridded Dataset is used. The manuscript presents original science that fits into the scope of GMD. The Introduction and the desciption of data and methods are especially well written. The discussion of the results and the presentation of the main findings and conclusions in the last section, however, fall a little bit short.

Specific comments:

**RC2:** In their description of the results, the authors often use terms like "significant gains", "significant percentages", "significant added value", "more/less significant" and also "very significant". Could you please specify if all these uses of "significant" are to be understood as a purely subjective estimation by the authors, or is it meant that the results are significant relative to a specific, objective degree? Has the statistical significance of the results been computed with a certain reference? Or does it simply mean that the obtained values are "somewhat large"? If the latter is the case, I would suggest to rephrase accordingly, and not use the term "significant" at all, to avoid confusion with an objectively calculated significance.

**AC:** We thank the reviewer by raising this issue. In this paper we did not perform any significance test. Thus, the terms "significant" refers to large values. We decided to follow the suggestion and change to "higher", "larger", "notable", "noteworthy" and "noticeable".

**RC2:** I would suggest to consider phrasings like "larger/higher" or "lower" instead of "strong" and "weak gains".

**AC:** We thank the reviewer for pointing out this issue and changed the instances with strong/weak into larger/smaller or higher/lower.

**RC2:** Line 353 is unclear to me. Could you please rephrase or clarify?

**AC:** We thank the reviewer for this note. We decided to remove the sentence: "For the maximum and minimum temperature extremes, the results are more limited, namely for TASMIN."

**RC2:** line 376: "gains not as relevant". Could you please clarify the meaning of relevant here?

**AC:** In this sentence we meant that the models listed have more limited gains in comparison with those listed before. We decided to change the sentence to "On the contrary, the GCMs RCMs pairs that displayed gains not as relevant as those listed before, all display points with limited values and sometimes small losses for sites in the interior, thus lowering the joint performance.

**RC2:** I believe "Summary and conclusions" would be a better fitting title for the last section, as it begins with a summary of the study, and much of the discussion has been done in the results section already.

**AC:** We thank the reviewer for the suggestion and changed to "Summary and Conclusions"

**RC2:** line 436: I am not sure what is meant with "approaching ERA-Interim and CMIP5 GCMs". Could you please clarify?

**AC:** We thank the reviewer for noticing the issue and decided to remove this part from the text.

**RC2:** line 437: "uncertainty associated is higher": Could you please discuss a little bit further what kind of uncertainty is meant here, and what this uncertainty means for the results of the study? Can this uncertainty be quantified somehow?

**AC:** We thank the reviewer for the suggestion. To clarify we changed the statement to "Nonetheless, since unrealistic values may be created, the uncertainty associated to this second approach is higher." This uncertainty is related to the creation of spurious values by downscaling the precipitation field from the low resolution by interpolating it to a higher resolution, thus not considering the effect that topography has on precipitation and land-atmosphere feedbacks. We also added this explanation at the end of the previous sentence: ", not taking into account local feedback systems and the effect that higher resolution topography has on precipitation"

**RC2:** Line 441: It is stated that "lower-resolution models will shower higher DAV values". Should it not be "downscalings of lower-resolution models show higher DAV"? Apologies if I misunderstood. Could you please clarify?

**AC:** We thank the reviewer for noticing this issue. We decided to change these lines to:

"While the DAVs metric allows for quantification of the gains or losses by the downscaling of the global models, no relationship is found when the same RCM is forced by multiple GCMs. More importantly a strong connection is observed for high-resolution models driven by the same GCM. The performance of the GCM along the regions of lateral forcing for the EURO-CORDEX plays an important role in the ability of the RCMs to downscale precipitation. This study clearly shows that the gains obtained from the use of higher resolution RCMs are paramount, not only owed to finer details in the representation of variables by itself, but also due to the increased description of

orography, and land-ocean-atmosphere feedbacks, which all have important impacts on precipitation."

**RC2:** I would suggest dividing lines 431-446 into several paragraphs, for the sake of readability. It is difficult to derive the main conclusions from this one big paragraph, which is a mixture of discussions and conclusions. A clearer structure and a sharper rephrasing would help in driving home the main message of this study.

**AC:** We thank the reviewer for the suggestion and revised the paragraph structure.

**RC2:** Could you please consider discussing what this study means for future research in this field? I believe you present important results here, and there is a good opportunity to finish the manuscript with a statement on a larger scope, setting the results in context with the aim of the CORDEX initiative, and future efforts in regional climate modelling.

**AC:** We thank the reviewer for the suggestion and added the following text to the last paragraph.

"The added value associated to the higher resolution gives credence to the growing effort to perform increasingly higher resolution simulations up to convection permitting scales. However, the inter-model variability supports the need for a coordinated ensemble of simulations similar to the one of the CORDEX Flagship Pilot Study on "Convective phenomena at high resolution over Europe and the Mediterranean". Increasing resolution implies higher computational costs; thus, in the last years the CORDEX community identified as a major challenge the objective quantification of RCM added value in respect to the GCM forcing. Added value assessments will allow the detection of future model development needs."

I am looking forward to a revised version of this manuscript.

Please find my technical comments below.

Technical comments:

**RC2:** Please carefully check the whole text for misplaced commas and the singular/plural use of verbs.

**AC:** We thank the reviewer for the suggestion and checked the whole text.

**RC2:** Throughout the manuscript there are several instances of the unit "km" being written as "Km". There is also often a space missing between number and "km".

**AC:** Corrected.

**RC2:** line 35: "assessment between" -> either "comparison between" or "assessment of"

**AC:** Corrected to "comparison between"

**RC2:** line 36: "Global" -> "global" and use of "PDF", an acronym which has not been introduced yet

**AC:** Corrected.

**RC2:** line 43: "become" -> either "has become" or "became"

**AC:** Corrected to "…has become…".

**RC2:** Throughout the manuscript: The term should called "convection permitting", and not "convective permitting". I would also recommend to use either British English or American English throughout the whole manuscript, instead of mixing the two. As for example British: "kilometre", "analysed" and American: "parametrization", "normalization", "~ized"

**AC:** We thank the reviewer for the suggestions and proceed to correct the whole manuscript with the proposed changes.

**RC2:** line 92: Maybe an article missing in front of "station-based dataset"?

**AC:** We thank the reviewer for the suggestion and added the following reference: "ECAD (Klein Tank et al 2002, Klok & Klein Tank 2009)

**RC2:** line 143: remove "is"

**AC:** Corrected.

**RC2:** line 144: remove comma after "one"

**AC:** Corrected.

**RC2:** line 172: "is" -> "are"

**AC:** Corrected.

**RC2:** line 173: "behaviour are" -> "behaviours are" or "behaviour is"

**AC:** Corrected.

**RC2:** line 210: "of" in front of "ERA-Interim"

**AC:** Corrected.

**RC2:** line 213, and other instances: "superior" -> "larger than" or something to similar effect
Throughout the manuscript there are many instances in which there is an unnecessary comma in front of the year in paranthesis, as for example in line 218: "Soares and Cardoso, (2018)".

**AC:** We thank the reviewer for raising this issue and corrected accordingly.

**RC2:** line 228: Is there something missing between "which" and "display"?

**AC:** Corrected. We changed "…from which display…" to "…of which they display…"

**RC2:** line 235: "can't" -> "cannot"

**AC:** Corrected.

**RC2:** line 242: "smoothing precipitation field" -> "smoothing of the … "

**AC:** Corrected.

**RC2:** line 288: "The next section" -> "This section", no?

**AC:** Corrected.

**RC2:** line 294: "was" -> "were"

**AC:** Corrected.